# HitNet: Hybrid Ternary Recurrent Neural Network

**Peiqi Wang**[1,2,4], **Xinfeng Xie**[4], **Lei Deng**[4], **Guoqi Li**[3], **Dongsheng Wang**[1,2*], **Yuan Xie**[4]

[1]Department of Computer Science and Technology, Tsinghua University
[2]Beijing National Research Center for Information Science and Technology
[3]Department of Precision Instrument, Tsinghua University
[4]Department of Electrical and Computer Engineering, University of California, Santa Barbara

wpq14@mails.tsinghua.edu.cn, {liguoqi, wds}@mail.tsinghua.edu.cn
{xinfeng, leideng, yuanxie}@ucsb.edu

## Abstract

Quantization is a promising technique to reduce the model size, memory footprint, and computational cost of neural networks for the employment on embedded devices with limited resources. Although quantization has achieved impressive success in convolutional neural networks (CNNs), it still suffers from large accuracy degradation on recurrent neural networks (RNNs), especially in the extremely low-bit cases. In this paper, we first investigate the accuracy degradation of RNNs under different quantization schemes and visualize the distribution of tensor values in the full precision models. Our observation reveals that due to the different distributions of weights and activations, different quantization methods should be used for each part. Accordingly, we propose HitNet, a hybrid ternary RNN, which bridges the accuracy gap between the full precision model and the quantized model with ternary weights and activations. In HitNet, we develop a hybrid quantization method to quantize weights and activations. Moreover, we introduce a sloping factor into the activation functions to address the error-sensitive problem, further closing the mentioned accuracy gap. We test our method on typical RNN models, such as Long-Short-Term Memory (LSTM) and Gated Recurrent Unit (GRU). Overall, HitNet can quantize RNN models into ternary values of {-1, 0, 1} and significantly outperform the state-of-the-art methods towards extremely quantized RNNs. Specifically, we improve the perplexity per word (PPW) of a ternary LSTM on Penn Tree Bank (PTB) corpus from 126 to 110.3 and a ternary GRU from 142 to 113.5.

## 1 Introduction

Recurrent Neural Networks (RNNs) yield great results across many natural language processing applications, including speech recognition, machine translation, language modeling, and question answering [1, 2, 3, 4, 5]. The emergence of various RNN architectures, such as Long-Short-Term Memory (LSTM) [6] and Gated Recurrent Units (GRU) [7], has achieved state-of-the-art accuracy in these applications. In order to further improve the model accuracy, researchers always build deeper internal recurrence by increasing the number of recurrent sequence length, hidden units, or layers. However, these methods usually introduce large models resulting in practical hardness, especially on embedded devices due to inferior resources for computation and memory. Model compression is a promising method to alleviate aforementioned problems. There exist several techniques to reduce the amount of model weights, such as low-rank approximation [8], sparsity utilization [9, 10, 11], and

---

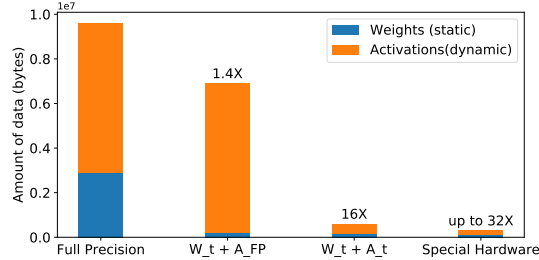

Figure 1: Theoretical memory saving for an LSTM model through quantization. $W\_t$ denotes ternary weights, $A\_t$ denotes ternary activations, and $A\_FP$ denotes full precision activations.

data quantization [12, 13]. Among these compression techniques, quantizing data into lower bitwidth is quite promising.

In previous studies that attempt to quantize RNNs to extreme low-bit precision, they always focus on weights while retaining activations in full precision [14, 15, 16]. Activations include all dynamic tensors during processing, and weights are learned and fixed after training. As shown in Figure 1, quantizing weights into ternary values {-1, 0, 1} can only save 1.4× memory consumption in the training phase. While quantizing both weights and activations can achieve up to 16× memory saving. Ternary values can be further compressed by using only 1 bit to represent -1 and +1 without storing the zero values on some specialized devices [17]. Memory requirements for this further compression can be reduced up to 32×. For the inference phase, although weights occupy the most of memory, the quantization of activations can further simplify the floating-point computations. Under the ternary scenario, the costly full precision matrix arithmetic can be transformed into simple logical operations, which are friendly for efficient implementations on customized hardware[18].

Quantization achieved impressive success and satisfied the accuracy requirement on convolutional neural networks (CNNs) even with only ternary weights and activations [19, 20, 21, 22, 23], while the results on RNNs are still unsatisfactory. The huge accuracy gap between the quantized models and the original models prevents RNNs from using extreme low-bit precision. This motivates us to investigate the reason for huge loss of accuracy in aggressive quantizations on RNNs. There are several differences between RNNs and CNNs. These differences could cause the practical success of state-of-the-art quantization methods on CNNs while similar methods cannot work well on RNNs. One major reason is the recurrent structure in RNN models. Activations in RNNs are iteratively updated along the temporal dimension through the same network structure and weights. This structure makes quantizing RNNs easier to cause an error accumulation. In addition, the strongly nonlinear activation functions (e.g. $sigmoid$ or $tanh$) in RNN are more sensitive to small errors compared to the piecewise linear functions in CNNs (e.g. $ReLU$). Although previous studies [14, 24, 25] tried to bridge this gap in various ways, the accuracy of quantized RNNs, especially the ones using extremely low-bit precision, is far from being satisfied.

In this paper, we analyze the value distributions of weights and activations to reveal the reason behind the accuracy degradation. Based on our observation, we propose a hybrid ternary quantization method called HitNet to significantly bridge the accuracy gap. In HitNet, we take full advantages of various quantization methods. Specifically, we use threshold ternary quantization method for weight quantization and Bernoulli ternary quantization method for activation quantization. Moreover, to further decrease the accuracy gap between the quantized models and the original models, we introduce a sloping factor into the activation functions to address the error-sensitive problem.

Our contributions can be summarized as follows:

- To the best of our knowledge, our work is the first to present a comprehensive analysis regarding the impact of different quantization methods on the accuracy of RNNs.

- Motivated by the above analysis, we propose a hybrid ternary quantization method, termed as HitNet, which adopts different quantization methods to quantize weights and activations according to their statistical characteristics.

- To further narrow the accuracy gap, we introduce a sloping factor into the activation functions. HitNet achieves significant accuracy improvement compared with previous work, like the

perplexity per word (PPW) of a ternary LSTM on Penn Tree Bank (PTB) corpus decreasing from 126 to 110.3 and a GRU from 142 to 113.5.

## 2 Related Work

Model compression is widely used to reduce the size of RNN models with redundancy, like low-rank approximation [8], sparsity utilization [9, 10, 11], and data quantization [12, 13]. All of these approaches are independent and we focus on data quantization in this work. The biggest challenge of quantization is to maintain application accuracy. Some methods [24, 26, 19] propose various transformation functions before uniform quantization or design different threshold decision approaches, trying to balance the distribution of quantized data and utilize limited quantized states efficiently. Zhou *et al.* [26] recursively partitioned the parameters by percentiles into balanced bins before quantization to balance data distribution. He *et al.* [24] introduced a parameter dependent adaptive threshold to take full advantage of the available parameter space. A few prior studies [14, 26] introduce preconditioned coefficients or employ various transformation functions before and after quantization to decrease the accuracy degradation. Some others formulated quantization into an optimization problem and then used different approaches to search optimal quantization coefficients, like greedy approximation [27] and alternating multi-bit quantization [25]. A few studies [16, 28] added extra components to reorganize the original RNN architecture and fine-tune the quantized models, but they introduce extra computation resulting in negative impacts on the performance for both inference and training phases. Most of these studies still present a big accuracy gap when using extreme low-bit precision. Our HitNet addresses this problem by quantizing both weights and activations of RNNs into ternary states with impressive accuracy improvement compared to previous work.

## 3 Analysis of Quantized RNNs

In this section, we provide a comprehensive study on the accuracy loss of quantized RNNs under different state-of-the-art quantization methods. We take LSTM as a case study to understand how these quantization methods work. The basic unit is the LSTM cell which can be described as

$$
\begin{aligned}
i_t, f_t, g_t, o_t &= \sigma(W_x x_t + b_x + W_h h_{t-1} + b_h) \\
c_t &= f_t * c_{t-1} + i_t * g_t \\
h_t &= o_t * \sigma(c_t)
\end{aligned}
\tag{1}
$$

where $\sigma$ denotes the activation function. The computation contains: input gate $i_t$, forget gate $f_t$, candidate cell state $g_t$, output gate $o_t$, cell state $c_t$, and hidden state $h_t$. The activation function is $\sigma(x) = sigmoid(x) = 1/(1+e^{-x})$ for $i_t$, $f_t$, and $o_t$, and $\sigma(x) = tanh(x) = (e^x - e^{-x})/(e^x + e^{-x})$ for $g_t$ and $h_t$. We refer to the first four items as gate computations in the following sections due to the similar formats. Weights in gate computations can be separated into two parts: one is related to current input information (i.e. $W_x x_t + b_x$) and the other is related to previous state (i.e. $W_h h_{t-1} + b_h$). Here $W_x$ ($W_h$) merges $W_{xi}$ ($W_{hi}$), $W_{xf}$ ($W_{hf}$), $W_{xg}$ ($W_{hg}$), and $W_{xo}$ ($W_{ho}$) for clarity. Activations represent the intermediate data propagated in the forward pass.

Existing quantization methods can be classified into three categories: uniformed quantization [26, 25], deterministic quantization (threshold-based quantization) [19, 21], and stochastic quantization [12, 13]. We use $Q(X)$ to represent the quantization method, where $X$ denotes the input tensor. Hence, the computation of its corresponding gradient in the back propagation is governed by

$$
\frac{\partial E}{\partial X} = \frac{\partial E}{\partial Q} \frac{\partial Q}{\partial X}
\tag{2}
$$

where $E$ denotes the loss function. For the non-differentiable part in $Q(X)$, we adopt the straight-through estimator (STE) method [29]. An STE actually can be regarded as an operator owning arbitrary forward operations but unit derivative in the backward pass. In the rest of this section, we will conduct detailed analysis for different quantization methods. All evaluations in this section adopt an LSTM model with one hidden layer of 300 units. The sequence length is set to 35, and it is applied on Penn Tree Bank (PTB) corpus [30]. The accuracy is measured in perplexity per word (PPW), and a lower value in PPW means a better accuracy.

Table 1: The testing perplexity per word (PPW) on quantized LSTM over PTB by using different quantization methods. $W\_2bit$ represents 2-bit weights, $W\_FP$ represents full precision weights, $W\_t$ represents ternary weights. The symbol $A$ for activations has the similar meaning. UQ denotes Uniform Quantization, TTQ denotes Threshold Ternary Quantization, and BTQ denotes Bernoulli Ternary Quantization.

<table>
<tr><td colspan="3" align="center">(a) UQ</td><td colspan="3" align="center">(b) TTQ</td><td colspan="3" align="center">(c) BTQ</td></tr>
<tr><td></td><td>$W\_2bit$</td><td>$W\_FP$</td><td></td><td>$W\_t$</td><td>$W\_FP$</td><td></td><td>$W\_t$</td><td>$W\_FP$</td></tr>
<tr><td>$A\_2bit$</td><td>638.9</td><td>108.9</td><td>$A\_t$</td><td>125.5</td><td>108.7</td><td>$A\_t$</td><td>189.5</td><td>108.3</td></tr>
<tr><td>$A\_FP$</td><td>103.3</td><td>97.2</td><td>$A\_FP$</td><td>98.6</td><td>97.2</td><td>$A\_FP$</td><td>110.9</td><td>97.2</td></tr>
</table>

## 3.1 Uniform Quantization

The basic and straightforward way to quantize a full precision value $x$ to a $k$-bit fixed-point one is the uniform quantization (UQ) [31], which can be implemented as

$$q_k(x) = \frac{round[(2^k - 1)x]}{2^k - 1}. \tag{3}$$

This quantization essentially reduces the representation precision rather than changing the dynamic range, thus $x$ needs to be mapped into [0,1] before quantization and then recovered back to the original range. Thus, the practical UQ is usually modified to

$$UQ(X) = 2max(|X|)[q_k(\frac{X}{2max(|X|)} + \frac{1}{2}) - \frac{1}{2}]. \tag{4}$$

In order to observe how UQ affects the accuracy of RNNs, we evaluate a 2-bit quantized LSTM on PTB and the results are shown in Table 1(a). We find that using 2-bit representation is relatively acceptable for just quantizing either weights or activations by UQ method. However, the accuracy degrades dramatically when we apply this quantization method on both weights and activations.

## 3.2 Threshold Ternary Quantization

Deterministic quantization assigns the quantized states arbitrarily according to an estimated threshold, and threshold ternary quantization (TTQ) [21] is such a method to convert full precision to ternary values. To retain as much information as possible, an estimated threshold $\theta$ and an non-negative scaling factor $\alpha$ are often used to minimize the quantization error. The TTQ method is implemented as

$$f(x) = \begin{cases} +1 & x > \theta \\ 0 & |x| \leq \theta \\ -1 & x < -\theta \end{cases} \tag{5}$$

$$TTQ(X) = \alpha f(X). \tag{6}$$

Assume that the data in RNNs obey a normal distribution $X \sim N(0, \sigma^2)$. We define a set $\Theta = \{i||x_i| > \theta\}$, and $I_\Theta$ be the indicator function of the set $\Theta$. Therefore, approximately solving an optimization problem of minimizing L2 distance between the original full-precision values and the quantized ternary values, we can get an sub-optimal threshold $\theta$ and the scaling factor $\alpha$ [21] calculated by

$$\theta \approx \frac{2}{3}E(|X|)$$
$$\alpha \approx \frac{1}{|\Theta|}\sum |x|I_\Theta(x). \tag{7}$$

We quantize both weights and activations in RNNs to ternary values using TTQ and the evaluation results are shown in Table 1(b). Quantizing weights has small accuracy loss compared to the original model while quantizing activations suffers from larger accuracy loss. Quantizing both of them also shows large accuracy degradation although it is much better than the result in the UQ case.

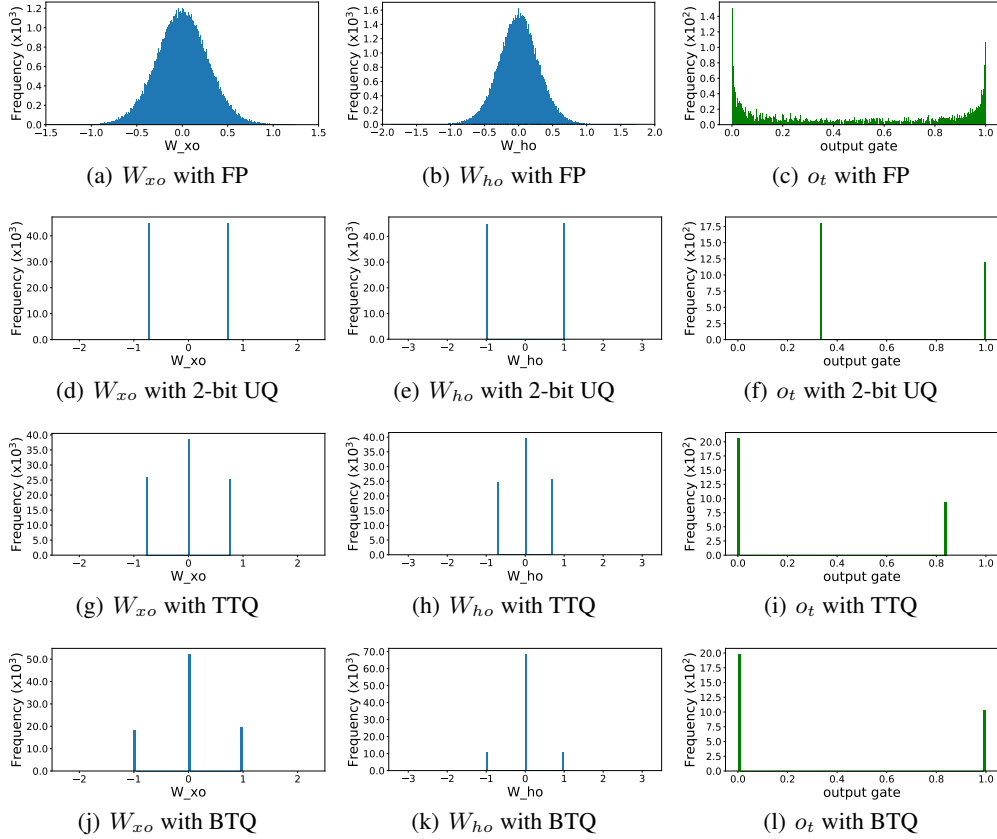

Figure 2: The distribution of weights (i.e. $W_{xo}$, $W_{ho}$) and activations of output gate (i.e. $o_t$) in LSTM under different quantization methods. (a)-(c) show the distributions in full precision (FP), (d)-(f) use the 2-bit uniform quantization (UQ), (g)-(i) use the threshold ternary quantization (TTQ), (j)-(l) use the Bernoulli ternary quantization (BTQ). The model used here is an LSTM trained on Penn Tree Bank (PTB) corpus.

### 3.3 Bernoulli Ternary Quantization

Stochastic quantization performs well in CNNs [13, 23]. Specifically, for the ternary case, the quantized value is sampled from a Bernoulli distribution. Thus, a Bernoulli ternary quantization (BTQ) is formulated as

$$f(x) = \begin{cases} +1 & \text{with probability } p \text{ if } x > 0 \\ 0 & \text{with probability } 1 - p \\ -1 & \text{with probability } p \text{ if } x < 0 \end{cases} \text{, where } p \sim Bernoulli(|x|) \qquad (8)$$

$$BTQ(X) = f(X). \qquad (9)$$

We also present the quantization test for BTQ, as shown in 1(c). For the weights outside the range of [-1,1], we apply a $tanh$ function for magnitude scaling before quantization [19]. We can see that the activation quantization obtains slightly better accuracy than quantizing weights. Quantizing both of them still suffers from unacceptable accuracy loss although it is also better than the UQ result.

### 3.4 Characteristic Analysis

In order to understand the accuracy gap between the original full precision and quantized low-bit precision with different quantization schemes, we first visualize the distributions of weights (i.e. $W_{xo}$ and $W_{ho}$) and activations (i.e. $o_t$) in Figure 2. Here we just take output gate as an analysis example. The distributions of full-precision data are shown in Figure 2(a)-2(c), and the rest figures represent models using various aforementioned quantization methods. These distributions are consistent with

our prior knowledge in terms of weights. The distribution of weights in RNNs follows the normal distribution although the range of $W_{ho}$ is slightly different from $W_{xo}$, which is shown in Figure 2(a) and Figure 2(b). However, the distribution of activations is very different from weights. The activation range is limited within [0,1] due to the activation functions. More importantly, most of the values locate near to the two poles instead of the middle of this range.

The distributions of the quantized data under 2-bit UQ scheme are shown in Figure 2(d)-2(f). It is found that this quantization method does not fully utilize the representation ability of 2 bits with 4 states and most values in this tensor concentrated to 2 states due to the unbalanced data distribution (other 2 states are too small to see in the figures). For quantized activations shown in Figure 2(f), only 2 states are valid due to the activation functions. This greatly degrades the model's expressive power. Some previous studies [26, 32] try to balance the distribution of quantized values to sufficiently utilize these 4 states for better representation. Usually, they introduce new transformation functions before mapping them into a 2-bit value to force them into a balanced distribution. However, such methods change the original data distribution to some extent, resulting in an unavoidable accuracy gap between the quantized model and the original model.

Figure 2(g)-2(i) show the distributions of quantized data under TTQ scheme, which provides good explanation for the comparable accuracy of just quantizing the weights. First, the distribution of quantized weights is similar to the original normal distribution. Second, all 3 states are fully utilized in the weight quantization. However, for the activation quantization, TTQ doesn't present good bipolar distribution as the original FP data. Therefore, although TTQ achieves better accuracy than the UQ, it still degrades the accuracy obviously when quantizing both weights and activations. Note that there are only 2 states in quantized activations, this is because we take $o_t$ with only positive values as the example, which is similar to the following BTQ scheme.

Figure 2(j)-2(l) present the distributions under BTQ scheme. Different from TTQ, activations in BTQ present bipolar distribution similar to the original FP data. Through probabilistic sampling, it performs a better approximation of the middle data in the stochastic sense. However, although the range of values in weights is preserved, 3 ternary states are not balanced since the lack of global statistic information (e.g. $\theta$ and $\alpha$ in TTQ). Therefore, BTQ scheme still suffers from similar accuracy loss when quantizing both the weights and activations.

In summary, these distributions under different quantization schemes provide insightful observations:

- TTQ is preferable for weights. Among these three quantization schemes, the TTQ utilizes 3 states efficiently and presents a balanced distribution.
- BTQ approximates the bipolar distribution of the activations better, and the probabilistic sampling preserves the middle data in the stochastic sense.

Although both of them still suffer from obvious accuracy degradation, the above comprehensive analysis motivates us to propose a hybrid ternary quantization method that takes full advantages of the different quantization schemes.

## 4 HitNet

Based on the above comprehensive comparisons, we propose HitNet in this section. We first introduce a constant sloping factor into activation to guide activations towards two poles, and then apply a hybrid ternary quantization method. Specifically, we use TTQ to quantize weights and BTQ to quantize activations to fully leverage the advantages of both. The hybrid quantization is governed by

$$
\begin{aligned}
i_t, f_t, g_t, o_t &= \sigma(TTQ(W_x)x_t + TTQ(b_x) + TTQ(W_h)h_{t-1} + TTQ(b_h)) \\
c_t &= f_t * c_{t-1} + i_t * g_t \\
h_t &= BTQ(o_t * \sigma(c_t))
\end{aligned}
\tag{10}
$$

We quantize the weights and hidden states since they are the only part involved in costly matrix operations, and the remained lightweight vector operations can be kept in high precision. In the training phase, the weights can be quantized right after each gradient step, while the activations need to be quantized every time the activation computation occurs. The threshold $\theta$ and the scaling factor $\alpha$ in TTQ are updated according to the real value distribution of the data to be quantized.

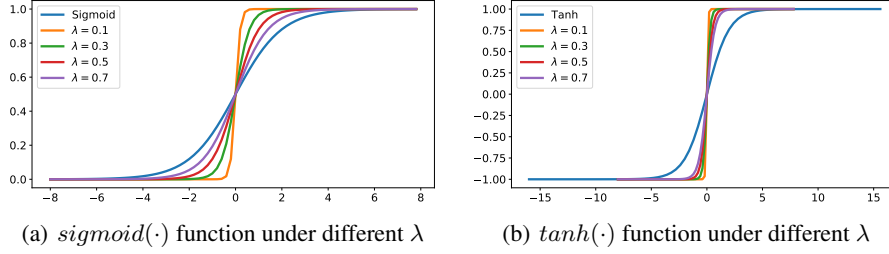

(a) $sigmoid(\cdot)$ function under different $\lambda$    (b) $tanh(\cdot)$ function under different $\lambda$

Figure 3: Activation functions under different $\lambda$.

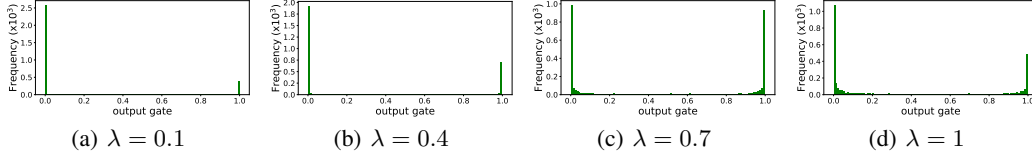

(a) $\lambda = 0.1$    (b) $\lambda = 0.4$    (c) $\lambda = 0.7$    (d) $\lambda = 1$

Figure 4: The distribution of output gate in LSTM with different sloping factor $\lambda$.

Because the activations present not only bipolar distribution but also with many middle data, none of the prior methods can avoid significant accuracy loss in an extreme low-bits quantization. The middle data between the two poles also play a non-negligible role on the model accuracy. If we can control the bipolarity of the activation distribution, we may have an opportunity to reduce the quantization error. Fortunately, prior work [33] proposed a temperature factor for the $sigmoid(\cdot)$ function in deep belief networks to control the activation distribution for better accuracy. It is also applicable to RNNs since the similar formats of activation functions. Therefore, we introduce a constant sloping factor in the activation functions to guide the activation distribution concentrating to the two poles as much as possible. Furthermore, based on previous analysis, we propose a hybrid ternary quantization method that uses TTQ for weight quantization and BTQ for activation quantization.

### 4.1 Hybrid Ternary Quantization

We introduce a constant sloping factor $\lambda$ to RNN cells for controlling the activation distribution:

$$\hat{sigmoid}(x) = \frac{1}{1 + e^{-\frac{x}{\lambda}}}, \quad \hat{tanh}(x) = \frac{e^{\frac{x}{\lambda}} - e^{-\frac{x}{\lambda}}}{e^{\frac{x}{\lambda}} + e^{-\frac{x}{\lambda}}}. \tag{11}$$

Figure 3 shows the variation tendency of activation functions under different $\lambda$. As prior work[33] pointed out that a properly small $\lambda$ ($\lambda < 1$) can scale the propagated gradients thus solve the gradient vanish problem to some extent to improve the accuracy; whereas, too small $\lambda$ will lead to extreme bipolar activation distribution that harms the model expressive power. Actually, a smaller $\lambda$ can also produce more bipolar activation distribution in RNNs as shown in Figure 4. This will reduce the middle data, and consequently improve the ternarization error from BTQ. Therefore, it is possible to obtain better accuracy on the quantized models if we carefully configure a properly small $\lambda$. We apply the modified activation functions during both the training and inference phases and evidence this prediction through the experimental results in Table 2. We use the same LSTM model as the one used in Section 3 for comparing the existing quantization schemes. The accuracy under different $\lambda$ for the full-precision model, quantized model, and the gap between them are clearly shown in the table. We get the following observations:

- A properly small $\lambda$ (0.3-0.6) can slightly improve the accuracy of the full-precision model, while too small $\lambda$ (<0.2) will obviously degrade the model functionality.

- A smaller $\lambda$ generates more bipolar activation distribution that reduces the ternarization error. It usually brings better accuracy on the quantized model except for smaller $\lambda$ with much worse baseline model in full precision. Jointly considering these two points, we configure $\lambda$=0.3-0.4 in this network.

Table 2: The testing PPW of LSTM models on PTB with different sloping factor $\lambda$.

| $\lambda$ | 0.1 | 0.2 | **0.3** | **0.4** | 0.5 | 0.6 | 0.7 | 0.8 | 0.9 | 1 |
|---|---|---|---|---|---|---|---|---|---|---|
| Full precision | 123.5 | 100.1 | **96.4** | **96.3** | 95.8 | 96.7 | 97.1 | 97.1 | 97.1 | 97.2 |
| HitNet | 136.4 | 115.7 | **110.6** | **110.3** | 112.8 | 114.6 | 115.6 | 116.3 | 119.9 | 120.1 |
| PPW degradation | -12.9 | -15.6 | -14.2 | -14 | -17 | -17.9 | -18.5 | -19.2 | -22.8 | -22.9 |

Table 3: The testing PPW of HitNet on the PTB dataset.

| Models | Kapur *et al.*[28] | He *et al.*[24] | Zhou *et al.*[26] | **HitNet(ours)** | FP(baseline) |
|---|---|---|---|---|---|
| LSTM | 152.2 | 152 | 126 | **110.3** | 97.2 |
| GRU | N/A | 150 | 142 | **113.5** | 102.7 |

- Actually, our hybrid ternary quantization method using TTQ for weights and BTQ for activations can improve the accuracy of the quantized model even without introducing the sloping factor (i.e. $\lambda$=1) compared to previous schemes. This can be seen if we combine Table 2 and Table 1.

In conclusion, our HitNet tries to find the best quantization strategies for different components in RNN models benefit from the hybrid quantization paradigm. Besides, the introduced sloping factor can guide the activation distribution more bipolar without affecting the accuracy of the full-precision model (actually slight improvement) and bridges the accuracy gap between the full-precision model and the quantized model. For the rest of the conducted experiments, we adopt the sloping factor $\lambda$ with the best accuracy (i.e. $\lambda$=0.4).

## 4.2 Overall Results

In this section, we evaluate the effectiveness of the proposed HitNet on language modeling with two typical RNN models (LSTM [6] and GRU [7]). The target is to predict the next word, thus we use perplexity per word (PPW) to measure the accuracy. A lower PPW means a better accuracy. For all RNN models in our experiments, a word embedding layer is used on the input side. We initialize the learning rate as 20 and decrease it by a factor of 4 at the epoch if the validation error exceeds current best record. The sequence length is set to 35 and the gradient norm is clipped into the range of [-0.25, 0.25]. In addition, we set the maximum epoch to be 40 and set dropout rate to be 0.2. The baseline of our experiment does not catch up with the state-of-the-art result, which is not within the scope of this paper. It can be further improved by optimizing the hyper-parameter configuration.

We first use the Penn Tree Bank (PTB) corpus [30], which contains 10K vocabulary. We conduct experiments for both LSTM and GRU, and provide comparisons to prior work. For a fair comparison, we use the model with one hidden layer of 300 hidden units, which is the same with previous counterparts. We use a batch size of 20 for training, and the results are shown in Table 3. We take the accuracy of 2-bit quantized models in previous work for comparison. In prior work [28], the result on 2-bit quantized GRU on PTB is not reported, thus we leave it. Although there is still an accuracy gap between our HitNet and the original full-precision model, HitNet outperforms existing quantization methods significantly. Furthermore, the 2-bit quantization in previous work actually has 4 states, while HitNet just has 3 states. Compared to these existing studies, HitNet achieves much better accuracy even using one less state for quantizing both the weights and activations.

We also apply our model to other larger datasets. Wikidata-2 is a larger corpus owning 33K vocabulary. It is roughly 2x larger in dataset size and 3x larger in vocabulary than PTB. We train both LSTM and GRU with one 512-size hidden layer and set the batch size to 50. Text8 is also a large corpus which is composed of a pre-processed version of the first 100 million characters from Wikipedia dump. We follow the same setting in [34] and the resulting vocabulary size is about 44K. On this dataset, we train the models with one hidden 1024-size layer and set the batch size to be 50. The results are shown in Table 4. HitNet is still able to achieve acceptable results on these larger datasets. We didn't compare the results with previous work because few of them focused on extremely low-bit quantization on these datasets.

Table 4: The testing PPW of HitNet on Wiki-2 and Text8 datasets.

| Dataset | LSTM | | GRU | |
|---------|------|------|------|------|
| | HitNet | Full Precision | HitNet | Full Precision |
| Wikidata-2 | 126.72 | 114.37 | 132.49 | 124.5 |
| Text8 | 169.1 | 151.4 | 172.6 | 158.2 |

Compared to prior work, HitNet demonstrates several advantages:

- HitNet adopts hybrid quantization methods for different data types to achieve better accuracy, providing a good match between the data distribution and the quantization scheme.
- HitNet introduces a sloping factor that can control the activation distribution to reduce the ternarization error.
- HitNet uses only 3 data states (less than 2 bits) that results in significant memory reduction via data compression and speedup via transforming full-precision matrix arithmetics to simple logic operations. Actually, the 0 values in HitNet do not need computation and can be skipped for further efficiency optimization on specialized platforms [9, 35, 36].

## 5   Conclusion and Future Work

We introduce HitNet, a hybrid ternary quantization method on RNNs to significantly bridge the accuracy gap between the full-precision model and the quantized model, outperforming existing quantization methods. The reason why previous extremely low-bit quantization on RNNs fails is comprehensively analyzed. Our observation reveals that the distributions of weights and activations are completely different. Prior work considers them equally leading to huge accuracy degradation in low-bit quantization. In HitNet, we quantize RNN models into ternary values {-1, 0, 1} by using TTQ (threshold ternary quantization) for weight quantization and BTQ (Bernoulli ternary quantization) for activation quantization. In addition, a sloping factor is introduced into the activation functions, which guides the activation distribution more bipolar and further reduces the ternarization error. Our experiments on both LSTM and GRU models over PTB, Wikidata-2, and Text8 datasets demonstrate that HitNet can achieve significantly better accuracy compared to previous work, and resulting in >16x memory saving via only 3 data states and extreme compute simplification via only logic operations.

In future work, we aim to deploy HitNet on specialized hardware to get practical acceleration and efficiency improvement because many characteristics of HitNet can be leveraged to obtain benefits. For example, the full-precision data can be compressed to only 2 bits or less with >16x memory saving; all matrix arithmetics are converted into logic operations, which can accelerate the execution and save the energy consumption; all zero values do not need to be stored and computed for further optimization.

## Acknowledgement

The authors were supported by the Beijing National Research Center for Information Science and Technology, the Beijing Innovation Center for Future Chip, and the Scalable Energy-efficient Architecture Lab (SEAL) in UCSB. This research was supported in part by the National Key Research and Development Plan of China under Grant No. 2016YFB1000303, the National Science Foundations(NSF) under Grant No. 1725447 and 1730309, and the National Natural Science Foundation of China under Grant No. 61876215. This work was also supported in part by a grant from the China Scholarship Council.

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
