[Reviews · NeurIPS 2018]

Reviewer 1



The authors study the problem of quantizing recurrent neural networks. While extreme low bit quantization (2 bits quantization) has achieved strong results for CNN, so far, such quantization performed poorly for recurrent neural network. The goal of this paper is thus to identify the reason for this observation, and to propose extreme quantization scheme better suited for RNNs. First, the authors compare different weight quantization: 2-bits uniform quantization, thresholded ternary quantization (TTQ) and Bernoulli ternary quantization (BTQ). This comparison is performed using a RNN trained on Penn TreeBank. The authors also report the distribution of the full precision and quantized weights of the network, as well as activations. Based on these experiments, they conclude that different quantization scheme should be used for weights and activation, as they have different distribution (gaussian for the weights, bimodal for the activation). Additionally, they propose to a "slop factor" to make the activation distribution more peaky, and thus further reduce the quantization error. They finally compare these approach, improving the perplexity on PTB from 126 to 110 (for ternary quantized RNN), while a full precision model gets 97. They also report results on wikitext-2 and text8, without comparison to previous quantization techniques. I liked the way that this paper is organized, starting from empirical observations to motivate the technical choices of the method. The problem studied in this paper is well motivated, and the paper is well written and easy to follow. My main concern regarding the writing of the paper is Figure 1: why is the memory used by activations so large, compared to the weights of the model? During inference, it is often not necessary to store all the activation, and since the motivation is to run the models on small devices, it probably does not make sense to use minibatch. Could the authors give more details about how these numbers were obtained? Regarding the training of quantized networks, how are the quantities for uniform and threshold quantizations (e.g. max, mean values) computed? Are they updated after each gradient step? Similarly, in the experiments reported in Table 2, is the slop factor applied at train time or only at test time? This paper is technically sound, and the proposed method is simple. However, I feel that the technical contributions are a bit incremental, as they are a fairly straightforward combination of existing techniques (threshold/Bernoulli quantization, and slop factor). For this reason, I believe that this paper is borderline for acceptance to the NIPS conference.

Reviewer 2



The author propose threshold ternary and Bernoulli ternary quantization methods for weights and activations, respectively. The paper sounds clear and the topic is quite significant. However, the originality (or contribution) sounds not enough, since they propose a ternary quantization which is in the middle of binary quantization and other quantizations, although they use hybrid and a sloping factor in addition to the ternary quantization. And, there are a few questions. - How about increasing the number of states rather than just 3 states? Can we find the best number of states in the trade-off between efficiency and accuracy? - For quantization of the weights, Wx and Wh are quantized. Then, how about activations. Actually, the gate computation includes 3 gates and 1 cell state, and the cell state might have a very different distribution than gates like the output gate in Figure 2. In addition, to be more detail, where are the quantizations applied in Eq. (1)? - In Eq. (7). during training, \theata and \alpha would change (be updated), which means activation functions are different for every iteration. Would it be ok, theoretically?

Reviewer 3



This paper introduces a variation on standard RNN formulations named HitNet. HitNet improves the SOTA of quantization performance for standard RNNs such as LSTM/GRU. Quantization has the benefit of allowing RNNs to operate with vastly reduced memory and computation requirements, but generally introduces a heavy penalty in terms of the relevant loss function (in this case, perplexity). The authors note that weights and activations in for LSTMs have hugely different distributions (approximately zero mean Gaussian for weights, whereas output activations are within [0,1] with most of the mass being at the extremes of the range). Quantisation for only one of weights or activations, whilst computing the other at full precision, produces a modest decrease in performance, but quantising both at once introduces a heavy penalty. Taking three standard quantization methods (Uniform Quanttization, Threshold Ternary Quantization, and Bernoulli Ternary Quantization) the authors make a compelling case that TTQ is preferred for quantizing weights, whereas BTQ is better for activations, due to the different distributions above. UQ is shown to be a poor proxy for both types of value, even though it has 4 possible values rather than 3. After the analysis, the authors introduce slope adjustable versions of standard activation function sigmoid and tanh. The sigmoid version was present in previous RBM work cited, whereas the tanh version is (to my knowledge) novel. The temperature parameter lambda controls the slope of the activation, and the authors demonstrate that sweeping this for sigmoid produces more polarised output distributions for an LSTM. By using lambda < 1, the output activations are closer to bipolar, and as such quantization produces less of a performance penalty. The authors choose a range for lambda by sweeping on both a full precision and a quantised model, finding a range of values which performs well for both. HitNet is comprised of the modified activations functions at the relevant places inside LSTM/GRU, as well as BTQ for activations and TTQ for weights. The authors evaluate on PTB, Wiki-2 and Text8. Significant improvements in perplexity are recorded compared to many other methods of quantizing models for PTB. For the larger datasets, competing benchmarks are not available, but HitNet seems to perform relatively close to the full precision models. Overall I find this paper very strong. The motivation is clear, the methodology is simply explained and seems easily reproducable. All experiments are described with sufficient hyperparameters, and the improvements over SOTA are significant. I recommend accepting to NIPS. Minor points: L120: "The recurrent timestep is set to 35" - I believe this refers to "sequence length" which I would find clearer. L139: I find the set notation difficult to understand here - should this be "the set of all i, such that |x_i| > \theta"? A diagram of the modified sigmoid and tanh activation functions, for a range of lambdas, would be an easily added element to aid comprehension.